# Not Every Image is Worth a Thousand Words: Quantifying Originality in Stable Diffusion

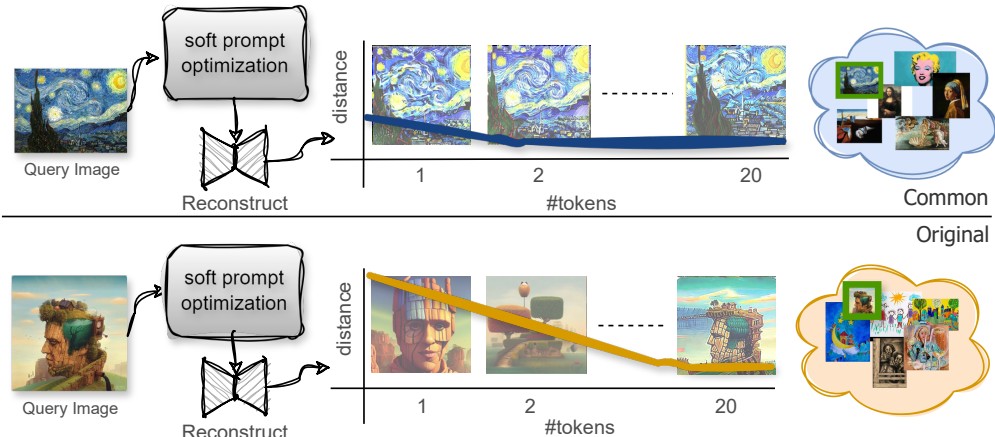

Figure 1: Illustration of our approach for measuring image originality. Using an off-the-shelf soft prompt method for images Gal et al. (2022), we reconstruct the image, and measure its quality. We argue original images require more tokens for accurate reconstruction, while common images like Van Gogh's "Starry Night" need only one token.

## Abstract

This work addresses the challenge of quantifying originality in text-to-image (T2I) generative diffusion models, with a focus on copyright originality. We begin by evaluating T2I models' ability to innovate and generalize through controlled experiments, revealing that stable diffusion models can effectively recreate unseen elements with sufficiently diverse training data. Then, our key insight is that concepts and combinations of image elements the model is familiar with, and saw more during training, are more concisely represented in the model's latent space. We hence propose a method that leverages textual inversion to measure the originality of an image based on the number of tokens required for its reconstruction. We demonstrate our method using both a pre-trained stable diffusion model and one trained on a synthetic dataset, showing a correlation between the number of tokens and image originality. Our approach is inspired by legal definitions of originality and aims to assess the trained model, without relying on specific prompts or having access to the training data. This work contributes to the understanding of originality in generative models and has implications for copyright infringement cases.

## 1 Introduction

Large-scale Text-to-Image (T2I) Generative Diffusion-based models have revolutionized our ability to produce visual content. T2I models, as their name suggests, are designed to produce images given a textual prompt. Distinctively from a search engine, these models are not meant to retrieve an existing image, but rather *generate* novel content that fits the description of the text. Hence, as the outcomes of the model generally did not exist before, quantifying generated originality remains a formidable challenge in practice and theory.

This challenge is not solely scholastic, and arises in the context of legal concerns surrounding copyright laws, where T2I models, trained on expansive datasets (Schuhmann et al., 2022) that include copyrighted materials, are often at the center of infringement accusations. Here too, quantifying originality poses a challenge as copyright law only protects the aspects of expressive works deemed *original* by the judiciary (Harper & Row, Publishers, Inc. v. Nation Enterprises, 1985; Feist Publications, 1991; U.S.C, 1990), where originality necessitates a minimal degree of creativity and authorship (Feist Publications, 1991).

In turn, methodologically sound methods for demonstrating creativity and originality in a T2I model become a pressing matter. Traditional strategies often formalize the problem of not-copying as a form of memorization constraint that inhibits overfitting of the data (Carlini et al., 2023; Bousquet et al., 2020; Vyas et al., 2023). This is also highlighted in the recently implemented EU AI Act, which mandates the disclosure of training data (Institute for Information Law (IViR), 2023), imposing difficult transparency in the operation and training of these models. However, regulating memorization is not necessarily aligned with the purpose of copyright law (Elkin-Koren et al., 2023), can be overly restrictive, and also poses computational, practical, and statistical challenges (Feldman, 2020; Feldman & Zhang, 2020; Attias et al., 2024; Livni, 2024; Zhang et al., 2016).

In this paper we consider an alternative viewpoint. Instead of looking at the training data and what information it holds, we analyze the model itself and what it had actually learned from the information the data has to offer. Specifically, We investigate whether T2I models can themselves be utilized to discriminate between generic and original content, according to their understanding of the world.

Toward this goal, we propose a quantitative framework to evaluate image originality based on the model's familiarity with training data. Our framework, tested on synthetic and real-world data, demonstrates T2I models' potential in identifying originality, helping develop metrics for auditing generative models and analyzing image originality.

We begin by assessing how well T2I models can innovate and generalize in a controlled set of experiments using synthetic data. Experiments to assess the generalization capabilities of generative models have been conducted in previous work (Zhao et al., 2018; Okawa et al., 2024), but the effects of textual conditioning have yet to be explored. As we demonstrate through a series of experiments, textual conditioning can help deepen our understanding of generalization. Our experiments reveal that stable diffusion models are particularly adept at recreating unseen elements when sufficiently diversified data is used. Overall, our findings underscore demonstrate prompted generations can creatively combine seen elements alongside unimaginative reconstruction, and underscore the importance of diversity in datasets (Lemley & Casey, 2020).

Next, we introduce our conceptual framework to quantitatively measure originality or genericity of images, followed by a practical implementation of it. Inspired by the theoretical work of Scheffler et al. (2022), we look at the complexity of description as a measure of originality. The working hypothesis is that common concepts are easier to describe in the machine's language (i.e., the latent space) than original concepts. Scheffler et al. (2022) build on the notion of Kolmogorov complexity. Similarly, we observe that the latent representation's length is also a great evaluator for complexity. We accordingly search for the shortest latent representation, using recent literature (Gal et al., 2022). By applying textual inversion techniques, we evaluate the extent to which a concept is familiar to the model, and thus, potentially unoriginal.

Finally, we validate our framework with empirical experiments utilizing both a widely used pre-trained stable diffusion model and a custom-trained model designed specifically for this study. The latter processes synthetic data composed of various shapes, colors, sizes, and infills. The experiments employ both textual inversion and DreamSim (Fu et al., 2023) to analyze the correlation between the ease of concept recreation — measured by the number of tokens needed — and the originality of the images relative to the training dataset . Our experiments reaffirm that embracing rather than avoiding memorization might enable generative models to produce more innovative and diverse content.

Overall we contribute to the study of originality and copyright in generative models by suggesting a new technique to identify genericity without imposing transparency, as well as offering an analysis in lab conditions that further our understanding of these models and how they behave.

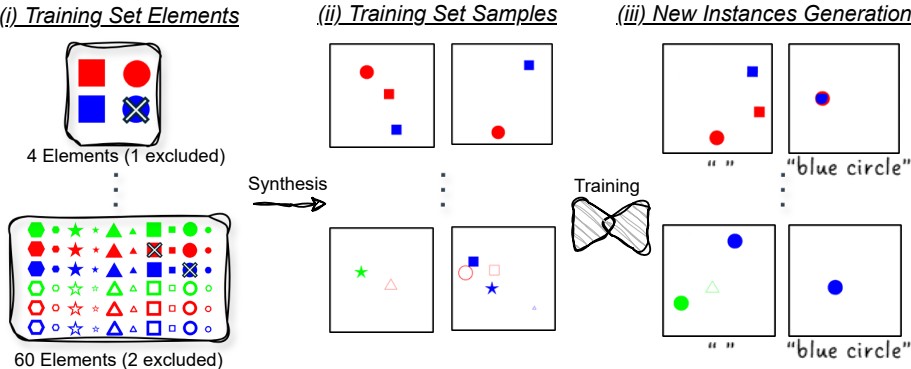

Figure 2: Generalization experiments diagram on synthetic data. **(i)** Training elements. We examine the relationship between data diversity and originality by running experiments over sets with distinct elements in increasing variety, with one or two elements excluded each time. **(ii)** For each dataset, we synthesis $100,000$ images, each consisting of $2-4$ random elements. **(iii)** T2I models are trained from scratch using the corresponding datasets. Left: examples of empty prompt generations. Right: examples of generations with the missing element as a prompt.

## 2 T2I MODELS PRODUCE ORIGINAL CONTENT

Before presenting our general framework, we first conduct preliminary experiments to establish T2I models ability to generalize and generate original content. Such experiments are prerequisite to any attempt to quantify such originality. The generalization abilities of generative models have been explored prior to the rise in T2I models popularity (Zhao et al., 2018), and in a simpler setting on diffusion models as well (Okawa et al., 2024). However, the impact of textual conditioning through prompts and the influence of training data diversity remain unexplored. In this section, we demonstrate, using the synthetic setup, that while diffusion T2I models obviously can memorize details from the training data and generate copied versions of it in the output, they can also generalize to new concepts and content through the composition of seen properties, surprisingly well. We then explore the ability of these models to generalize, considering the influence of the training data distribution. Specifically, we examine how the diversity of the data and the guidance provided by prompts contribute to the models' generalization capabilities.

We introduce a generalization assessment setup, as depicted in fig. 2, and present experimental findings in fig. 3 and expand on those in the supplementary. Our quantitative analysis reveals that generalization improves with increased training data diversity and textual conditioning. Additionally, we observe an enhancement in the quality of generated images with greater training data diversity.

**Setup and Methodology** The experiment evaluates the model's ability to generalize by withholding specific elements during training and assessing their generation post-training. Each element in the dataset has four dimensions: Size, Color, Texture, and Shape Type. The dataset's diversity ranges from minimal, with two shape types (square and circle) and two colors (red and blue), to maximal, with five shape types (square, circle, triangle, hexagon, and star), three colors (red, green, and blue), two sizes (big and small), and two textures (full and empty), creating 60 unique elements. The degree of generalization is quantified by the frequency of the missing element's occurrence in the generated set.

This experiment is conducted twice: using an empty prompt and with a prompt describing the missing element. Results are averaged over multiple experiments with different spanning sets and missing elements. An illustration of the experiment is provided in fig. 2, and additional details on the synthetic framework setup and methodology are provided in the supplementary.

**Data Diversity Promotes Generalizability** The synthetic experiments yield evidence that, indeed, the models are capable of generalizing and generating novel content. Results are summarized in

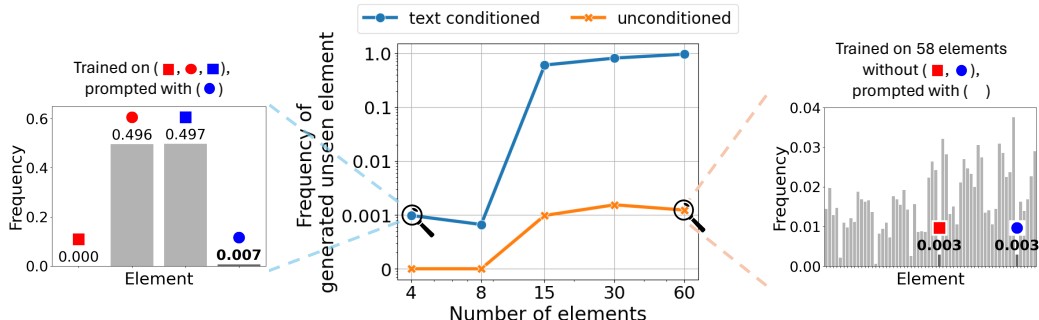

Figure 3: Synthetic generalization experiments results. **Center:** Generalization capability of the trained models vs. training data diversity (x-axis) and conditioning types (blue line vs. orange line). **Sides:** Detailed distributions for a specific set and missing elements. These results support the notion that models generate both original and reproduced content, highly depending on the training data.

Fig. 3, which depicts how diversity in the training data enhances generalization. Prompting allows us to further exemplify this by actively requesting new unobserved content.

Our results show that increasing training data diversity helps the model to generalize. By prompting a request to an element not seen during training, we can see that a dataset containing as few as 60 unique elements yields the requested element consistently. This demonstrates the model's ability to deconstruct and reconstruct elements and effectively translate them between textual and visual domains. When the model is trained on monotonous datasets, namely datasets with relatively few elements, the model collapses into the behavior of copying and fails to reconstruct novel elements. In a typical example, in the text-conditioned simple model, a model that was trained on blue squares and red circles was prompted with "blue circle". The model generated images with two elements, a blue square *and* red circle, but failed to generate the novel concept of blue circle. Interestingly, the quality and expressiveness seem to correlate. This is illustrated in fig. 2(iii).

Quality improves with training data diversity alongside the generalization frequency. For example, comparing the generated blue circles of the simple model with those of the diverse model, the diverse model consistently produces higher-quality results, regardless of conditioning type. This trend is consistent across various elements, as demonstrated in the supplementary. These observations highlight the significance of dataset composition in fostering model creativity and robustness.

Overall, these experiments establish models do not just memorize the data but can also deconstruct elements and concepts and reconstruct them in creative ways; we can, therefore, now proceed to measure originality with such models.

## 3 MEASURING ORIGINALITY USING CONDITIONED TEXT

Before presenting our method, we provide a background overview for some key-ingredient methods that we use, namely Stable diffusion Rombach et al. (2022) as our T2I models architecture, and a textual inversion Gal et al. (2022) as the process for measuring complexity.

**Stable Diffusion** As this model has gained high familiarity in the community, we provide a short description of its components. The Stable Diffusion model generates images conditioned on textual input. The image generation process involves encoding the input image into a latent representation using a *Variational Autoencoder (VAE)*, and then refining this representation using a *U-Net*, conditioned by a text embedding from the *Text Encoder*. Finally, the refined representation is decoded back into an image using the VAE decoder, resulting in an output image that is both realistic and semantically aligned with the input text.

**Textual Inversion** Textual inversion is a method employed in T2I latent diffusion models, such as Stable Diffusion, to adapt the model for generating images that are specific to a particular visual concept or object, which may not have been present in the original training data. This is achieved by fine-tuning a pre-trained T2I model on a selected set of images $x_1, x_2, \ldots, x_n$ that represents the target concept. The fine-tuning process results in the creation of a distinctive token $S^*$, which

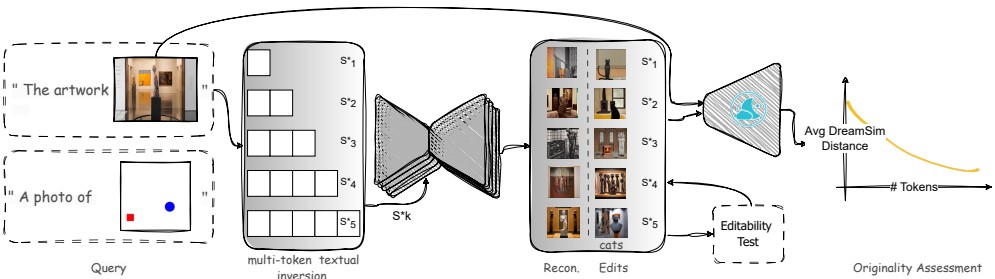

Figure 4: Method overview. We begin with a query image and a domain-relevant prompt (left). The query is processed through textual inversion Gal et al. (2022) with different token lengths. With each inversion, images are reconstructed and edited (generation with variations). After ensuring each reconstruction is in-distribution, we estimate the concept generative quality Fu et al. (2023) (right).

encapsulates the visual characteristics of the concept. During the training process, only the parameters associated with the text embeddings are updated, while the parameters of the VAE and the U-Net components of the model remain unchanged. This selective training approach ensures that the model retains its general image generation capabilities while learning to associate the new token $S^*$ with the specific visual attributes of the concept. Once trained, the token $S^*$ can be used in the text input of the T2I model to generate new images that exhibit the learned concept, allowing for controlled and targeted image synthesis.

### 3.1 METHOD

The overview of our approach can be seen in fig. 4, essentially performing image reconstruction and quality assessment. Our approach builds on the textual inversion technique (Gal et al., 2022) for reconstruction, originally designed for personalization and editing tasks by representing concepts with a textual embedding. Unlike the original purpose of the method, our research aims to enhance the interpretability of the manifold of text-to-image (T2I) models, focusing on the originality of images rather than objects. For this purpose, we extend the method to employ *multiple tokens* instead of just one, building on the fact that a single-token representation may not sufficiently capture a complex, original, image. Then, we assess the generation quality in terms of reconstruction and editability. Overall, we find and demonstrate that the number of tokens required for reconstruction is correlated with the originality of an image.

**Single Token vs. Multi-Token:**  Let $T$ be a set of tokens representing a concept, where $T = t_1, t_2, \ldots, t_m$ is a multi-token representation with $m$ tokens. In the original textual inversion method, a single token ($m = 1$) is used to represent a concept. In contrast, our extension allows for multiple tokens ($m > 1$) to represent the concept in a more detailed manner. The representation of the concept in the latent space can be expressed as a sequence of the embeddings of the tokens: $S_m^* = e_t(t_1)e_t(t_2), \ldots, e_t(t_m)$, where $e_t$ is the embedding function and $S_m^*$, is the concatenated embedding of the tokens representing the query image. This multi-token approach enables a more granular exploration of the T2I model's manifold, facilitating a deeper understanding of the relationships between text and image representations, especially in the context of interpreting the originality of individual images. Following the creation of $S_m^*$, we can then use the original model to query with this new sequence $S_m^*$ using the existing vocabulary of the text encoder. Once we have the $S_m^*$ sequence that represents the query image, the overall process involves two main steps: reconstruction and in-distribution evaluation.

**Reconstruction Quality**  To assess the quality of reconstruction, we employ the DreamSim score by Fu et al. (2023), a SOTA distance metric to measure the similarity between the generated image and the query image. For a given image represented by a set of tokens $T = t_1, t_2, \ldots, t_n$, we generate a set of images $x_1', x_2', \ldots, x_{20}'$, where each image $x_i'$ is generated using the textual inversion method with tokens $T$. The reconstruction score for each image is calculated as: Reconstruction Score$(x_i') =$ DreamSim$(x_i', x)$, where $x$ is the original query image. The overall reconstruction score for the concept is the average of the scores for the 20 generated images: Average Reconstruction Score$(T) =$

$\frac{1}{20} \sum_{i=1}^{20}$ Reconstruction Score$(x_i')$. Lower scores indicate better reconstruction and the results are plotted for visualization to provide a comprehensive understanding of the model's performance. Other distance metrics, such as L2 and LPIPS, were explored but yielded significantly poorer results. Further details on the DreamSim metric are provided in the supplementary material.

**In-Distribution Assessment** We employ our method in two experimental setups, one synthetic playground for controlled occurrence distributions and their induced behavior and the other for the real-world scenario. For *Real-world* Settings, we use editability as the criterion to maintain in-domain generation Gal et al. (2022; 2023). Specifically, we use prompts like "cat in $S_m^*$" to generate images that are both representative of the concept in the query image and editable within the domain of the model. For the *synthetic* settings, due to the model's simplicity of the data it was trained on, editability is not necessarily the right measure of in-domain generation. However, The underlying advantage of the synthetic data is that the distribution of the data is fully known. This allows us to measure in-domain generation by a more informed measure which we can validate. Instead of editability, we check that for every seed, the shapes are in different positions, as the data distribution positioned the shapes randomly (by design). This validation ensures that the model has not reached a point of overfitting. We provide an ablation study to justify this decision in the supplementary.

Finally, we assess the originality of the query images by combining the reconstruction and in-distribution validation. The combination of multi-token textual inversion and these evaluation criteria enables a more detailed and original content generation, contributing to the assessment of the originality of imagery and Interpretability of T2I diffusion models.

## 4 EXPERIMENTAL SETUP

We conduct our experiments on two main environments. Both employ the stable diffusion architecture and differ in the data they were trained on. The first controlled environment, is trained from scratch on synthetic data, detailed in section 2. The second is the common pretrained stable diffusion [1]. Additional analysis is provided in the supplementary material. In the synthetic setting, we include a study on in-distribution assessment and offer a cleaner in-depth evaluation of the originality of the generated shapes. For the real-world setting, we present an additional experiment, on images that are guaranteed not to be in the training data, demonstrating that few-token reconstructions are not the result of duplication in the training data, thereby differentiating between popularity and lack of originality.

### 4.1 SYNTHETIC FRAMEWORK

Our first set of experiments is conducted on the synthetic dataset described in section 2. As discussed, within this synthetic framework we are able to provide evidence for generalization and hence validate the assumptions underlying our method for quantifying originality. To conduct the experiment, separate Stable Diffusion models were trained on synthetic datasets as depicted in section 2. We use a VAE and U-Net from scratch and employ BERT as the Text-Encoder. For evaluation, a YOLOv8 model is fine-tuned on the synthetic datasets to detect and classify elements in generated images, ensuring high-quality detections with a confidence threshold of 0.9. Additional details are described in the supplementary.

**Quantifying Originality in Synthetic Framework** In the context of assessing the originality of query images, we synthesize a custom dataset characterized by a non-trivial distribution. This dataset features three distinct element combinations, varying in occurrence frequency within the dataset, differing by orders of magnitude. Notably , genericity is simple to quantify in this setting, as it is the occurrence frequency. We validate our originality quantification methodology within this controlled setting.

---

[1]we based our implementation on the Huggingface Diffusers framework at `https://huggingface.co/docs/diffusers/en/index`, and used its pretrained stable diffusion models.

| Domain | Initial Token | Common | | | Original | | |
|--------|---------------|--------|---|---|----------|---|---|
| Houses | `"house"` | | | | | | |
| Artwork | `"painting"` | | | | | | |
| Animals | `"animal"` | | | | | | |
| Sport Photography | `"sport"` | | | | | | |
| People | `"person"` | | | | | | |

Table 1: Sample Images from our dataset. We curated 20 images for each of the 5 presented categories. In each category, 10 images are common, and 10 are original according to a legal expert. Samples are provided for each category.

## 4.2 REAL-WORLD SETTING

In the more elaborate setting, we used the widespread public Stable diffusion model[2]. We demonstrate our method on diverse domains, including houses, artworks, sports, animals and human faces. For each, we initialize all tokens in the learned prompt with a relevant initial token, train the model to discover tokens for the query image, and measure editability by validating the existence of a cat when prompted with "cat in $S_m^*$.". A List of the domains evaluated and examples of the curated data is provided in table 1. For each domain, we use 20 images (10 original and 10 common), manually selected from the web by a legal expert, resulting in a total dataset of 100 images. The dataset will be publicly released to support future research.

**Implementation Details**   We trained a multi-token textual inversion variant with different sizes of token length, starting from 1 and up to 5 consecutive tokens in the sequence. The training was conducted with a batch size of 20, a learning rate of $5e - 4$, and a total of 2000 steps, using 50 denoising inference steps. Further details, including the training prompts and the training scheme, are provided in the supplementary.

## 5 RESULTS

In this section, we present the results of our experiments (see settings in section 4). Additional Results, including demonstrations of the In-Distribution assessment, are provided in the supplementary.

**Synthetic experiments**   The results of the synthetic experiments are summarized in fig. 5. On the right side of the figure, we present the quantitative results of the experiment validating our originality measurement method. In this analysis, we randomly selected 20 images with concepts from each one of the following prevalence categories: Common, which makes up 30% of the data; Rare, which represents just 0.1% of the data; and Unseen. The plot summarizes the minimum number of tokens required for successful reconstruction for these 60 images. A concept is a combination of two shapes (e.g., a circle and a square) comprising an image. The results reveal that the majority of Common images can be reconstructed with just one token, while Rare images typically demand 2 to 3 tokens. Unseen images generally require 4 to 5 tokens, and sometimes even more (marked by "+" in the plot).

---

[2]The pretrain Stable diffusion model was taken from `https://huggingface.co/CompVis/stable-diffusion-v1-4`

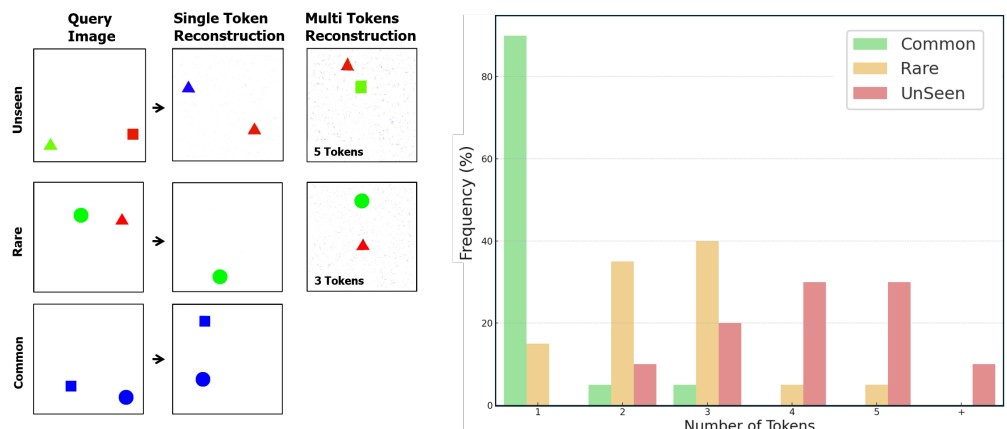

Figure 5: Minimum tokens required for image reconstruction based on data frequency. The right plot shows the distribution curves around the number of tokens needed for Common, Rare, and Unseen images. The left side presents qualitative results, demonstrating that Common images require 1 token, Rare images require 3 tokens, and Unseen images require 5 tokens for accurate reconstruction.

In addition, we provide qualitative instances from this experiment on the left side of fig. 5. Familiar concepts (dubbed as "common") that are well-known to the model, require only a single token for accurate reconstruction (bottom). In contrast, rare examples necessitate three tokens (middle), while unseen concepts, which the model has never encountered before, require five tokens for correct reconstruction (top).

This experiment highlights the varying sequence lengths necessary for reconstructing images across different categories, validating the significance of token count in reflecting the model's familiarity with a concept.

**Pretrained Stable Diffusion**    Qualitative examples for reconstruction using textual inversion for original images (as labeled by a human legal expert) are provided in fig. 7 and for common images in fig. 6. Familiar identities (e.g., Barack Obama, a common cat) and common image settings (e.g., a frontal view of a house or a team standing together) are well reconstructed, even with the use of a single token. However, for images labeled as original (e.g., a green cat, or an unfamiliar human), a higher number of tokens is required.

Quantitative results for all images collected in table 1, as detailed in section 4.2, are plotted in fig. 8. From the results, it's evident that multiple tokens are necessary for accurate reconstruction of original images. Overall, semantic preservation improves with the addition of more tokens for original content, while it is already high on the first token for common content.

|  | Houses | Art | Sport Photography | Animals | People |
|---|---|---|---|---|---|
| Query Image |  |  |  |  |  |
| Single Token Reconstruction | 0.31 | 0.39 | 0.29 | 0.28 | 0.43 |

Figure 6: Qualitative results for reconstructing common images from our domains using single-token textual inversion. As can be seen, for common images a single token reconstruction reaches high reconstruction similarity. The average DreamSim score for each experiment is depicted at the bottom of each representative image.

| Domain | Query Image | 1 token | 2 tokens | 3 tokens | 4 tokens | 5 tokens |
|--------|-------------|---------|----------|----------|----------|----------|
| Houses | | 0.71 | 0.48 | 0.56 | 0.38 | 0.32 |
| Art | | 0.63 | 0.53 | 0.57 | 0.52 | 0.52 |
| Sport photography | | 0.77 | 0.36 | 0.54 | 0.56 | 0.43 |
| Animals | | 0.6 | 0.48 | 0.41 | 0.33 | 0.52 |
| People | | 0.7 | 0.6 | 0.83 | 0.7 | 0.39 |

Figure 7: Qualitative results for reconstructing original images from our domains using multi-token textual inversion. As can be seen, for original images more tokens improve capturing additional details of the query image. The average DreamSim score for each experiment is depicted at the bottom of each representative image.

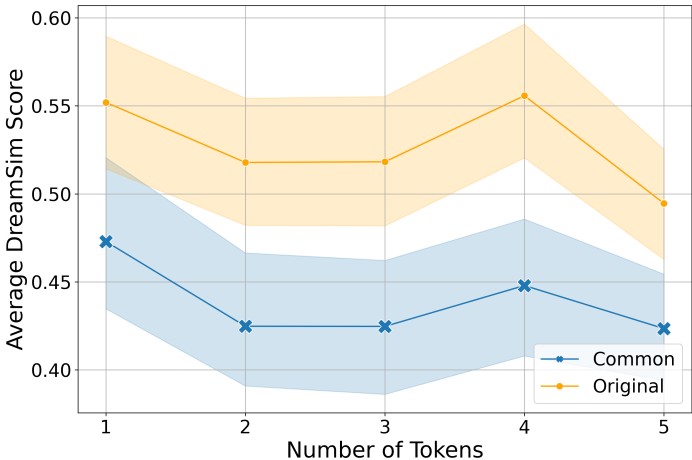

Figure 8: Quantitative results show that increasing the number of tokens in multi-token textual inversion improves the ability to capture semantic details of the query image for original images. However, for common images, even a single token achieves strong reconstruction performance.

## 6 RELATED WORKS

**Privacy and Copyright Infringements** The intersection of privacy and copyright infringements in generative models has garnered significant attention. This approach assumes that to avoid copyright infringement, the output of a model shouldn't be too sensitive to any of its individual training samples. Bousquet et al. (2020) suggest the use of differential privacy Dwork et al. (2006) to stabilize the algorithm and avoid such sensitivity. Vyas et al. (2023) introduces a slightly less stringent notion (Near-Access Freenes) but relies on a similar benchmark, a *safe model* that doesn't have access to the copyrighted data. Nasr et al. (2023), Carlini et al. (2023) and Haim et al. (2022) further explored this area by investigating the extraction of training data from models, highlighting the risks of memorization. Elkin-Koren et al. (2023), however, investigated the gap between privacy and copyright infringement from the perspective of the law, and showed that requiring such notions of stability may be too strong, and are not always aligned with the original intention of the law.

Other work, including Lee et al. (2023) and Hacohen et al. (2024), try to highlight the challenges in copyright assessment in generative AI and how to use methods such as in-painting to discover biases of the models. Closer to our approach, Scheffler et al. (2022) suggests a framework to quantify originality by measuring the description length of a content with and without access to the allegedly copyrighted material. Our approach of textual inversion also looks for a succinct description of the content but, distinctively, our definition depends on the distribution of the data, and measures originality with respect to the whole data to be trained. This may lead to different outcomes, for example, when the allegedly copyrighted material contains a distinctive trait that is not necessarily original.

**Attribution in Generative Models**  Attribution in generative models is a crucial area of research, focusing on identifying the sources of data that contribute to the generation of specific outputs. Park et al. introduced the TRAK method to address data attribution in large-scale models (Park, 2022), and recently, Wang et al. (2023) proposed a method for evaluating the attribution in Stable Diffusion models of data points in the generation process, which is closely related to assessing the originality of generated images. However, such a method requires full access and knowledge of the training set on which the model was trained.

**Generalization and Memorization**  The interrelation between generalization and memorization is a key challenge for Machine Learning. Classically, memorization and generalization are considered to be in tension. Ideal learning would seem to *extract* relevant information but avoid memorizing irrelevant concepts. While limiting memorization does lead to generalization (Russo & Zou, 2019; Bassily et al., 2018; Xu & Raginsky, 2017; Arora et al., 2018), recent studies suggest that memorization may be critical, and unavoidable in certain tasks (Feldman, 2020; Feldman & Zhang, 2020; Livni, 2024). Most recently, (Attias et al., 2024) demonstrated how even in simple tasks such as mean estimation, memorization of the data is a prerequisite. On a practical level, (Zhang et al., 2016) explored the relationship between these two aspects, emphasizing their importance in the effectiveness of deep learning models.

# 7 DISCUSSION

In this work, we introduced a novel approach to assess the originality of images with Text-to-Image (T2I) Generative Diffusion models, and have investigated its behavior in this aspect under a controlled environment. Our methodology leverages the concept of familiarity within the model's training data to quantify the originality of tested images. By employing textual inversion techniques, we demonstrated that the number of tokens required to represent and reconstruct an image is a measure of its originality, without requiring access to the training data, nor a specific prompt that potentially poses copyright complications. Our analysis confirmed that T2I models can produce new original content, highlighting the importance of training models on diverse and comprehensive datasets. These findings also challenge the traditional view of avoiding memorization in models. Instead, we propose that models familiarize themselves with a broad spectrum of data, respecting copyright constraints, to enhance their ability to generate new content. In the supplementary we provide additional discussion on the impact and ethical aspects of the work. In summary, our study introduces a novel approach to evaluating originality in generative models, offering insights that inform copyright analysis and legal protection. By quantifying concept familiarity, we address issues of copyright eligibility, infringement, and licensing while also opening new research avenues in originality assessment.

**Limitations**  One of the primary constraints for the method is the reliance on textual inversion, which may not capture all aspects of originality in complex images. Additionally, our method's effectiveness is contingent on the quality and diversity of the training data, which might not always be optimal. Furthermore, the correlation between token count and originality, although significant, may not be universally applicable across different model architectures or datasets. Future research should explore alternative measures of originality and test the robustness of our approach across a broader range of models and data, making it readily available for deployment. Finally, our work demonstrates that T2I models can be utilized to discriminate original and non-original work. That being said, an important motivation of our work is to assess originality of T2I content. Designing a framework that exploits generative model's ability to discriminate original content in order to audit genAI and safeguard content leads to several open problems and challenges which we leave to future work.

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
