# Not Every Image is Worth a Thousand Words: Quantifying Originality in Stable Diffusion Supplementary Material

## 1 Additional Results

We include additional results for both Common and Original categories in each domain. Results for Houses, Sports Photography, Animals, Art, and People from the original set are shown in fig. 1, and from the common set in fig. 2.

Figure 1: Addtional Qualitative results demonstrate the effectiveness of multi-token textual inversion in reconstructing original images across different domains, with more tokens enhancing the capture of additional details. On the right, we demonstrate the editability test using the prompt "Cat with $S_m^*$" for $m = 4$.

Figure 2: Additional qualitative results show that multi-token textual inversion can effectively reconstruct common images from various domains, with a single token often sufficing for high-quality reconstruction. The editability test in the last row is illustrated using the prompt "cat with $S_m^*$".

## 2 SYNTHETIC FRAMEWORK

### 2.1 IMPLEMENTATION DETAILS

**Datasets**  All images feature a white background and a number of geometric shapes in the foreground. Each *element* in the image is defined by four features, and the entire set is the cross-product of all features. The shapes are independently and uniformly located across the image. The image's default textual description is in the format *"big red full circle and small empty blue square"* [1].

Utilizing this framework, we generate datasets comprising 100K images each. In every dataset, 10% of the images are empty, while the rest contain a variable number of elements uniformly distributed within the range $[1, n]$, where $n$ can be any natural number. In the following sections, we choose either $n = 4$ or $n = 6$.

**Models**  We train a separate Stable Diffusion model for each dataset, so that the only visual data seen by the resulting model is the training dataset itself. Key decisions included (i) pretraining a VAE (Encoder-Decoder) and a UNet (noise-cleaner) from scratch, and (ii) employing BERT as the Text-Encoder, chosen over CLIP to avoid the broader visual context implications associated with CLIP training.

**Evaluation**  In order to facilitate automated and large-scale analysis of generations, which is essential for the purposes outlined in Sec. 2 and 4.1, we fine-tuned a YOLOv8 model on the synthetic datasets. This approach effectively addresses common issues, such as overlapping or slightly deformed elements, providing a confidence measure for each detection. We set the confidence threshold at 0.9, aligning with the requisite quality of the generated elements.

### 2.2 GENERALIZATION EXPERIMENTS ADDITIONAL DETAILS

**Generalization target**  In assessing generalization, we leave out specific elements from the training process, and ask for their generation after training. While these elements have not been witnessed by the model, their properties have. For example, if a blue circle is omitted from training, the model still witnesses circles and the color blue, only not in conjunction (Fig. 2(i), bottom). The degree of generalization can be evaluated by the frequency of occurrence of the missing element within the generated set, normalized by the total number of generated elements. We repeat this experiment with a prompt asking for the specific missing element, and unconditionally with the empty prompt.

**Text conditioning**  In evaluating each trained model, we assess the occurrence frequency of the generated missing element across two sets, each comprising 1024 generated images. Initially, we generate images using an empty prompt (i.e., " "), thereby sampling from the unconditioned distribution represented by the model. Subsequently, we generate images with a prompt precisely describing the missing element (e.g., "blue circle"), thereby sampling from the model's text-conditioned distribution. It is natural to anticipate that employing a specific textual prompt will increase the frequency of the generated missing element.

**Training data diversity**  As discussed in the experiments section (sec. 4 of the main paper), each element within our dataset encompasses values of four dimensions: Size, Color, Texture, and Shape Type. Our dataset's diversity spectrum ranges from the least diverse, characterized by a span of two shape types (square and circle) and two colors (red and blue), as demonstrated in fig.3 (left) of the main paper, to the most diverse, which encompasses four dimensions - five shape types (square, circle, triangle, hexagon, and star), three colors (red, green, and blue), two sizes (big and small), and two textures (full and empty), thereby resulting in 60 unique elements (main paper, fig.3 right). Consistent with prior research Zhao et al. (2018), we anticipate a positive correlation between diversity and generalization.

---

[1]Based on empirical experiments, we have found this more effective than counting or grouping elements in the prompt. Evidence for misalignment between the prompt and produced images has been shown in various studies as well Chefer et al. (2023); Wang et al. (2023)

**Addressing Bias**    To mitigate potential bias arising from the model's inclination towards certain values within our element subspace, we enforce symmetry by averaging over a larger number of experiments, each differing in its spanning set and missing elements. For instance, the leftmost data points in fig.3 in the main paper, represent the averaged results of four identical 4-element experiments conducted sequentially with (1) big full elements, (2) big empty elements, (3) small full elements, and (4) small empty elements.

# 3    IN-DISTRIBUTION ASSESSMENT IN THE SYNTHETIC SETTING ABLATION STUDY

In Section 4 of the main paper, we outline simplified criteria for in-distribution testing within a synthetic setting vs a real-world setting, where we employ the concept of edibility as a metric. In the synthetic domain, a crucial test for the model's effectiveness lies in its ability to generate images without merely copying the spatial placement of elements from the query image. To test this, we provide the following ablation study, where we fixed the location of elements in common images during the training phase. The rationale behind this methodology was to challenge the notion that the model's generation of elements in varied locations might still be indicative of overfitting and to ensure that the model stays within the intended distribution bounds.

Given that our model operates on patches, it could be suggested that if the model recreates identical elements in different locations, it might not be exhibiting true understanding but rather a form of overfitting. To address this, we trained the model on images with a single fixed location, hypothesizing that if the model were able to replicate these elements in the same fixed location, it would demonstrate an awareness of element locations beyond mere memorization.

The results of the ablation study supported our premise: the elements from the common images consistently appeared in the same spatial positioning as in the query image, providing evidence of the model's spatial awareness. This finding is vital as it suggests that the model's generation of elements in different locations is not an artifact of overfitting but rather an indication of its genuine understanding of elements.

Qualitative illustrations from this study are presented in fig. 3, with the original query images on the left and the single-token reconstructions generated using four distinct random seeds on the right.

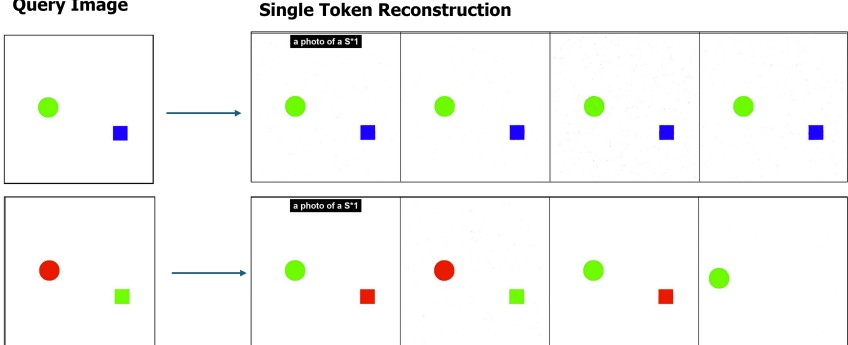

Figure 3: Qualitative Illustrations of the Ablation Study. The original query images on the left and their corresponding single-token reconstructions on the right, generated using four distinct random seeds. These examples serve to validate the model's capability to comprehend and maintain the fixed spatial locations of elements as observed in the common images during training, demonstrating the validity of our in-distribution test in the synthetic setting.

# 4 SYNTHETIC IMAGES QUALITY ANALYSIS

As illustrated in fig. 4, we observed an expected relationship between training data diversity and generated image quality. The quality of generated elements not present in the training set improves with greater diversity. Additionally, we found that the generation quality was not impacted by the type of conditioning, particularly in more diverse cases where unconditioned generation led to the creation of missing elements.

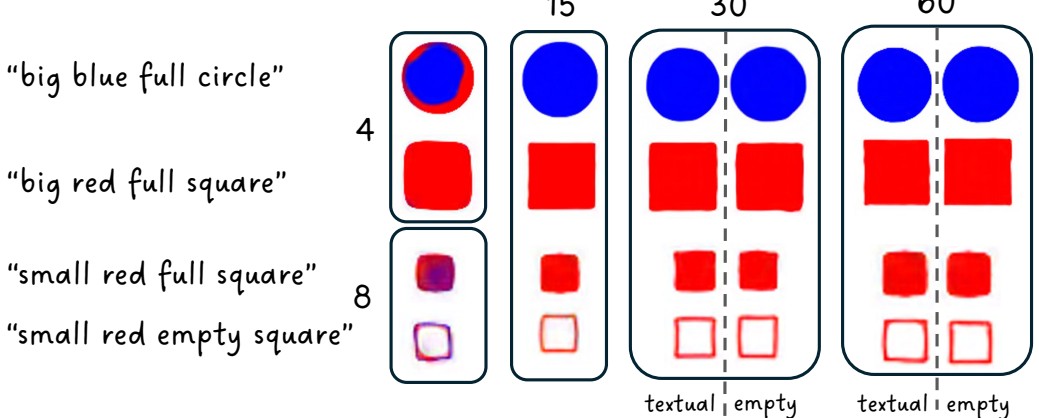

Figure 4: Detailed set of generated samples showcasing the correlation between training data diversity and generated images quality. **Rows:** All elements in a row are missing elements of the same class, when appearing in generated images. **Blocks:** All elements in a block where generated from models trained on data of the same diversity scale, ranging from 4 elements to 60. **Sub-blocks:** In the 30 and 60 blocks, the right-hand side sub-blocks represent the results of empty prompts, and the left-hand side sub-blocks represent the results of ad-hoc textual prompts.

In addition, we provide sample generations from each of the experiments in this study in fig. 5.

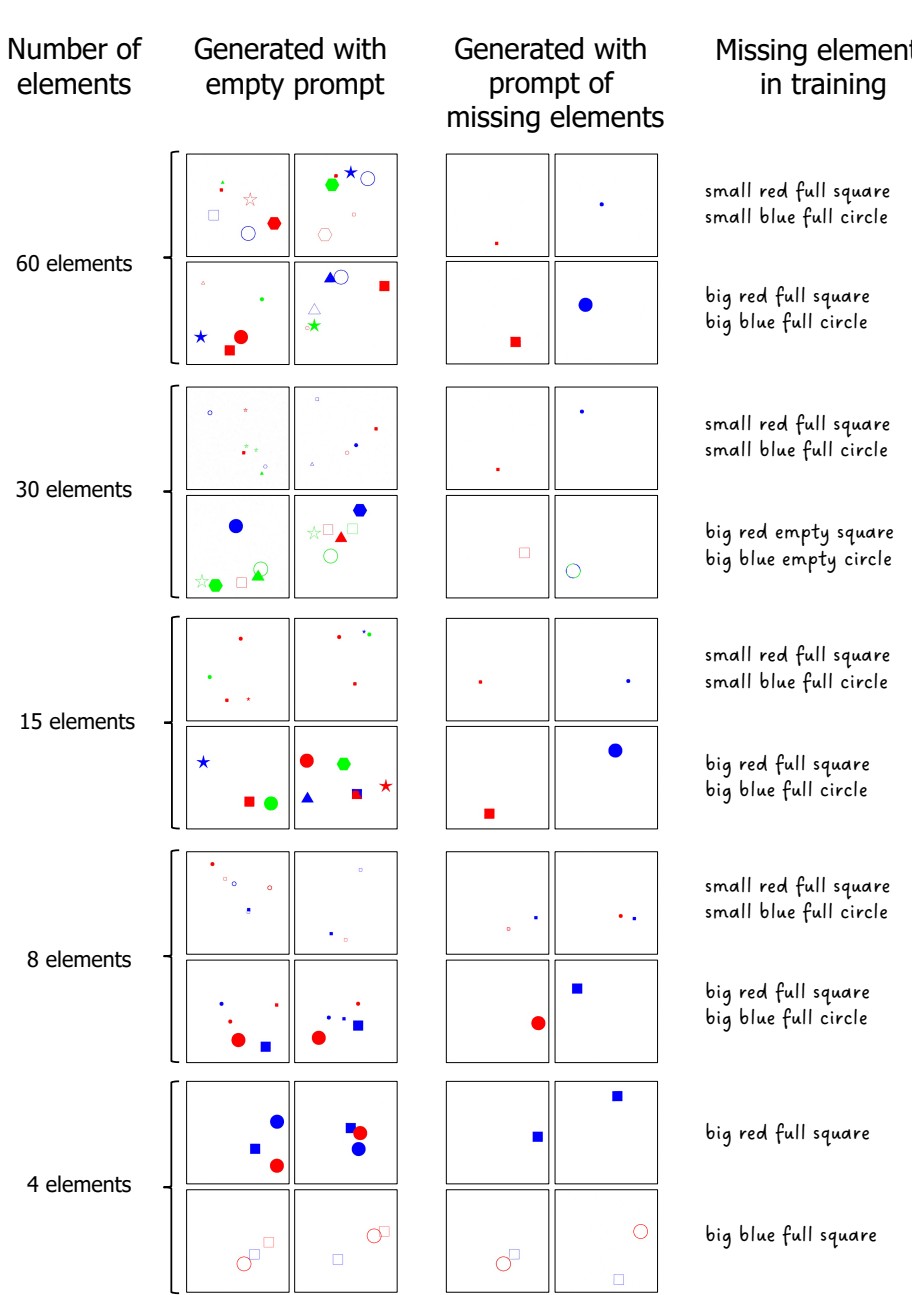

Figure 5: A representative sample of generations made by the synthetic models. Each row contains images generated by a different model. The models differ by the number of different element in their training set, and the specific elements left out. The left columns contain generations with an empty prompt, while the right columns contain images generated with a prompt describing one of the elements missing in training.

## 5 DISTINGUISHING BETWEEN POPULARITY OF IMAGES IN THE TRAINING DATA AND UNORIGINALITY

As the common images in Section 4.2 of the main paper were collected from the web, one could argue that they may be highly duplicated in the LAION database, which Stable Diffusion was trained on. Consequently, the phenomenon of low-token reconstruction might result from the model memorizing those specific images rather than recognizing the general concept.

To address this concern, we conducted an experiment using images of street cats, which are known for their generic appearance and are familiar to the model, but were taken by the authors of this paper. Thus, it is guaranteed that these images were not part of the training data.

We used a set of 15 images captured specifically for this experiment and applied our method as described in section 3 of the main paper. We found that the average DreamSim distance for all images with a single-token reconstruction was 0.42, consistent with the results for single-token reconstruction of common images in Section 5 of the main paper. Examples of single-token reconstructions for these images are provided in fig. 6. The full set of images will be made publicly available.

Figure 6: The top row shows the images taken by the authors, which were not part of the training data, ensuring no prior exposure to the model. The bottom row presents the reconstructions generated using a single token, demonstrating that generic concepts can be effectively reconstructed with minimal tokens, and this is not a result of memorization.

## 6 QUANTIFYING ORIGINALITY USING TEXTUAL INVERSION: ADDITIONAL IMPLEMENTATION DETAILS

This section provides detailed information on the implementation of the models described in Section 4 of the main paper.

### 6.1 SYNTHETIC EXPERIMENTS

**Stable Diffusion Pre-Training Details**   Our T2I model training involved two stages. First, we trained a VAE for 8 epochs with an effective batch size of 32 and a learning rate of $10^{-5}$. Next, we trained a UNet for 15 epochs using the trained VAE and a pre-trained BERT model, with an effective batch size of 64 and a learning rate of $6.4 \times 10^{-5}$.

We evaluated our method's originality assessment in a controlled environment by synthesizing a custom dataset with specified features: Type: [circle, square, triangle, hexagon, star], Color: [red, green, blue], Size: [big], Texture: [full]. The dataset was structured as follows:

- **Common:** 30% of the images contain the pair (circle, square) in varying colors.
- **Rare:** 0.1% contain the pair (circle, triangle) in varying colors.
- **Unseen:** The pair (square, triangle) does not appear in any images.

The remaining images may contain up to 4 elements, with no more than one from the set [circle, square, triangle]. We consider the frequent pair (circle, square) as generic, with multiple variations in each image's element positions. Our method quantifies originality based on this stable diffusion model, using a total of 100K instances for training.

**Multi-Tokens Textual Inversion Training Details**    In the synthetic setup, we trained our multi-token textual inversion variant with token lengths ranging from 1 to 5. The training used a batch size of 20, a learning rate of 0.0005, and 2000 steps. We found that 50 denoising inference steps produced cleaner results.

## 6.2 PRETRAINED STABLE DIFFUSION EXPERIMENTS

**Reconstruction Measurement**    We measure the similarity between original and reconstructed images using the DreamSim distance. DreamSim is built upon an ensemble of different models; for our use case, we used the DreamSim distance, which includes all models. At evaluation time we generate for each image 20 images using the prompt `"a photo of `$S_m^*$`"` and average the dreamsim score.

**Training Prompt Templates**    Similar to the original Textual Inversion method, we used object text templates for all experiments except the art domain, following the approach in [reference]. For the art domain, we used a custom list generated by GPT4 and manually curated. The full list of text templates includes:

- "a detailed image of the artwork titled $S_m^*$"
- "a high-resolution photo of the artwork $S_m^*$"
- "a close-up view of the artwork known as $S_m^*$"
- "a digital representation of the art piece $S_m^*$"
- "the famous artwork $S_m^*$"
- "a full view of the art piece titled $S_m^*$"
- "an artistic interpretation of $S_m^*$"
- "a gallery display of the artwork $S_m^*$"
- "a photographic capture of the art $S_m^*$"
- "the artwork $S_m^*$ in full detail"
- "a visual study of the artwork $S_m^*$"
- "the complete artwork known as $S_m^*$"
- "an exhibition view of $S_m^*$"
- "a curated image of the artwork $S_m^*$"
- "a detailed scan of $S_m^*$"
- "an artistic rendering of $S_m^*$"
- "a high-quality image of the artwork $S_m^*$"
- "the full artwork titled $S_m^*$"
- "a museum display of $S_m^*$"
- "an archival photograph of the artwork $S_m^*$"

## 7 DREAMSIM DISTANCE METRIC

In section 3 of the main paper, we describe our method for measuring originality, which includes measuring the distance between the query image and the reconstructed image. DreamSim is an advanced perceptual image similarity metric with STOA performances that offers a more comprehensive and human-aligned approach to evaluating image similarity compared to traditional methods like the Fréchet Inception Distance (FID).

DreamSim is designed to bridge the gap between low-level image metrics (such as LPIPS, PSNR, and SSIM) and high-level semantic judgments (such as those made by models like CLIP). Traditional metrics often fall short in capturing mid-level differences in image layout, object pose, and semantic content, which are crucial for aligning with human visual perception.

DreamSim leverages embeddings from several pre-trained models, including CLIP Radford et al. (2021), OpenCLIP Ilharco et al. (2021), and DINO Caron et al. (2021). These embeddings are fine-tuned using human perceptual judgments on a dataset of synthetic images created by text-to-image models. The fine-tuning process involves learning from around 20,000 image triplets, where human annotators have determined which images are more similar.

The formulation of DreamSim can be summarized as follows:

- *Concatenation and Fine-tuning:* The embeddings are concatenated and fine-tuned on human perceptual judgments: $E_{\text{concat}} = \text{concat}(E_{\text{CLIP}}, E_{\text{OpenCLIP}}, E_{\text{DINO}})$, $E_{\text{DreamSim}} = \text{fine-tune}(E_{\text{concat}}, \text{human\_judgments})$ Where, $(E_{\text{CLIP}}, E_{\text{OpenCLIP}}, E_{\text{DINO}})$ are the embedding functions of the respective models.

- *Cosine Similarity:* The perceptual distance $D$ between two images $I_1$ and $I_2$ is computed as the cosine distance between their embeddings:

$$D(I_1, I_2) = 1 - \frac{E_{\text{DreamSim}}(I_1) \cdot E_{\text{DreamSim}}(I_2)}{\|E_{\text{DreamSim}}(I_1)\| \|E_{\text{DreamSim}}(I_2)\|}$$

**Advantages over Traditional Metrics**

- *Human Alignment*: DreamSim is trained on human judgments, making its similarity assessments more aligned with how humans perceive visual similarity.

- *Comprehensive Feature Capture*: By using embeddings from multiple models, DreamSim captures a wide range of visual features, from low-level textures to high-level semantic content.

- *Generalization:* Despite being trained on synthetic data, DreamSim generalizes well to real images, making it versatile for various applications, including image retrieval and reconstruction tasks.

## IMPACT STATEMENT

This paper presents work whose goal is to advance the field of generative models by introducing a framework for quantifying originality in text-to-image diffusion models. The potential broader impact of this work includes the following:

**Ethical Aspects**  Our research addresses the challenge of quantifying originality, which has significant implications for copyright laws and the protection of creative works. By providing a methodology to assess the originality of generated images, we aim to contribute to a fairer and more transparent use of generative models in creative industries. This could help mitigate legal disputes related to copyright infringement and ensure that the rights of original content creators are respected.

**Future Societal Consequences**  The ability to quantify originality in generated images could enhance the deployment of generative models in various fields, including art, design, and entertainment, by fostering trust and accountability. It can also encourage the development of new creative tools that assist artists in generating unique content while respecting intellectual property rights.

Overall, we believe that our contributions to the understanding of originality and creativity in generative models will have a positive societal impact by promoting ethical use and fostering innovation.