# OpenReview forum: "Not Every Image is Worth a Thousand Words: Quantifying Originality  in Stable Diffusion"
_ICLR.cc/2025/Conference — ICLR 2025 Conference Withdrawn Submission_

### Official Review · Reviewer_8bKc · 2024-10-29

**Soundness:** 2
**Presentation:** 3
**Contribution:** 3
**Rating:** 5
**Confidence:** 3

**Summary:**

The paper studies the originality of generated content from text-conditional diffusion models. In particular, the authors find that the number of tokens required to invert a generated image is a strong indicator of originality. The authors posit that this work contributes towards the understanding of generated images’ originality and can be useful in the current discussions around copyright infringement from generative models.

**Strengths:**

Strengths:

- The paper is extremely well-written, the scientific hypothesis is backed by relevant literature and synthetic experiments for validation.
- The detection method is simple to use and is built upon existing works such as Textual Inversion — which can enable widespread adoption by the community. I also like that the idea is motivated by a simple observation of “description complexity” as a measure for originality.

**Weaknesses:**

Although the method is simple and has the potentially to be widely adopted, I believe the paper contains the following weakness:

- For a pre-trained model, it would be a good idea to relate the results to the pre-training data frequency for a particular concept. This can provide an extra layer of validation, as currently the ground-truth definition of “originality” is a little less principled. I can suggest a few ideas in this regard: The authors can compute a distance metric (e.g., DreamSim) between the query image and the training set images (e.g., LAION 400M which can be accessed) — and retrieve the top-k images. A diversity computation amongst the top-k images can give a signal towards originality. However, I believe there would be other ways to automatically obtain the ground-truth annotations for “originality”. In that case, the author’s method will be a strong motivation as it bypasses access to the pre-training corpus (which might not be accessible in a lot of cases).
- I find that the paper is lacking a bit on the copyright infringement legal discussion.  How can this method be useful for copyright infringement detection?  The authors can reference some recent works on this line: https://arxiv.org/abs/2404.08030, https://blog.genlaw.org/pdfs/genlaw_icml2024/55.pdf.
- While the paper curates a method — and baselines in this regard are difficult to find — The authors are suggested to at least provide a discussion around the missing baselines.

**Questions:**

See Weaknesses:

Overall, I believe the method can be useful — but the paper needs to be supplemented with solid applications using it (e.g., how can it be useful for copyright infringement detection).  I am happy to revisit my score after the discussion period if the authors address the Weakness points.

---

> ### Author Response · Authors · 2024-11-14
>
> We appreciate the reviewer's evaluation of our work and their constructive feedback.
>
> "For a pre-trained model..." - We agree that it would be valuable to measure the frequency of our images in the dataset. However, given the size of LAION, it is completely intractable to verify the frequency of a particular image, especially using metrics like DreamSim.
>
> "I find that the paper is lacking a bit on the copyright..." - see the answer to Reviewer k77t (2)

---

### Official Review · Reviewer_4K7W · 2024-11-01

**Soundness:** 2
**Presentation:** 3
**Contribution:** 3
**Rating:** 5
**Confidence:** 3

**Summary:**

This paper proposes a framework for evaluating the **originality** of text-to-image (T2I) generative models, particularly in the context of copyright considerations.
The authors focus on how originality can be empirically assessed through the analysis of latent representations of images produced by these models.
They argue that originality can be measured by the number of tokens required for a model to reconstruct an image, relating this to legal definitions of originality that emphasize a minimal degree of creativity and authorship.
The authors conduct controlled experiments that demonstrate T2I models’ capabilities to generate unseen elements when trained on diverse datasets.
The paper addresses the implications of their framework for copyright analysis, asserting that it can inform discussions about copyright eligibility, infringement, and licensing within AI-generated content.

**Strengths:**

## Strengths

1. Novel Concept: The method leverages the number of hidden states in a new and innovative way, presenting a fresh and interesting idea.
2. Clear Writing: The paper is well-organized and easy to understand, facilitating comprehension of the proposed concepts.

**Weaknesses:**

## Weaknesses

While the paper introduces some interesting ideas, there are several major issues:

1. Dependence on Soft Prompt Optimization. The method relies heavily on soft prompt optimization techniques, such as multiple tokens textual inversion. This reliance can lead to instability and slow performance. For example, generating the distance versus number of tokens curve (as Figure 1) for each image likely requires substantial computational resources, making the method impractical as a standard evaluation tool like FID.

2. Existing methods can reproduce the original image by selecting specific noise seeds. Several studies have explored this area [1,2], which raises questions about using the number of hidden states as a measure of originality. With effective seeds, the need for multiple hidden states is reduced.

3. The connection between "easier to describe" and "shorter token length" is tenuous and lacks (theoretical) support, making it difficult to justify the underlying assumptions of the method.

4. Even if the relationship between "easier to describe" and "shorter token length" holds, the model appears to measure ***relative originality/creativity*** [3]. This measure depends on the training dataset of the diffusion model, raising questions about its effectiveness for applications such as copyright infringement detection.

   - [1] Meiri, Barak, et al. "Fixed-point Inversion for Text-to-image diffusion models." *arXiv preprint arXiv:2312.12540* (2023).
   - [2] Qi, Zipeng, Lichen Bai, and Haoyi Xiong. "Not all noises are created equally: Diffusion noise selection and optimization." *arXiv preprint arXiv:2407.14041* (2024).
   - [3] Wang, Haonan, James Zou, Michael Mozer, Anirudh Goyal, Alex Lamb, Linjun Zhang, Weijie J. Su et al. "Can AI be as creative as humans?" *arXiv preprint arXiv:2401.01623* (2024).

**Questions:**

## Questions

1. If an image is associated with a long textual sequence containing special characters and appears multiple times in the training set, what outcome is expected? Does the method anticipate requiring multiple hidden states for such instances, or will it consolidate them into a single hidden state?

---

> ### Author Response · Authors · 2024-11-14
>
> We appreciate the reviewers' careful consideration of our work and the constructive feedback provided.
>
> Adressing the Weaknesses:
> (1) That is true, and we hope future work will lead to more scalable and computationally tractable methods. However, the use case we have in mind does not involve large-scale verification of originality. Instead, the method is designed for specific cases where the originality of a certain image, intended to be protected, needs to be evaluated. Such use cases do not necessarily require very fast or highly efficient methods and can afford to rely on a more computationally intensive processes.
>
> (2) Thank you, this is an interesting ablation study that we can conduct and include in the final version of this work.
>
> (3) Relying on short description length as a measure of originality has been discussed theoretically (see Schefer et al. in our references). But, since computing the description length is an intractable problem, the work presented here provides empirical (rather than theoretical) evidence for the applicability of this approach.
>
> (4) That is true, and our underlying assumption is that a model like SD has been trained on a sufficiently large and representative dataset that reflects the true underlying distribution. Again, we provide empirical evidence for this assumption in the paper.
>
>  As for the question:
>
> The working hypothesis of our study is that the description of an image, given prior or trained images, can be much shorter than the description of the image without knowledge of the distribution. This is precisely why our measure of originality is not based on the textual sequence used to generate the image, but rather the potential textual sequence derived from existing images in the training set. For example, the description of "Starry Night" might be quite long in isolation, but due to its high frequency in the training set, its description becomes much shorter when considering the distribution.

---

### Official Review · Reviewer_k77t · 2024-11-02

**Soundness:** 2
**Presentation:** 2
**Contribution:** 3
**Rating:** 3
**Confidence:** 4

**Summary:**

The paper presents a new technique for measuring the originality of an image leveraging diffusion models and textual inversion. Namely, originality is quantified by the number of tokens (learned via textual inversion of a diffusion model) needed so that reconstruction based on this learned sequence of tokens is of high quality. Alternatively, one could use the reconstruction error when restricted to learning a single token per image as a measure of originality.

The authors study their proposed measure of originality in two settings: (i) a synthetic one, where the training data is controlled, allowing for characterizing an image as 'common', 'rare', or 'unseen', and (ii) a 'real-world' setting consisting of 10 'common' and 10 'original' images for 5 broad categories. In both cases, the number of learned tokens needed is higher for the less common instances. The authors also present evidence they claim shows that diffusion models can create novel images consisting of an unseen combination of previously seen elements.

**Strengths:**

I like the **overall idea**! The notion that something more 'original' would be harder to reconstruct is sensible, and using the number of tokens needed for successful textual inversion is neat.

I appreciate the **experiments with synthetic data**. Relating model behaviors to training data distributions is often overlooked and I find these analyses to be quite insightful.

The result of **figure 5** offers evidence for the notion that required number of tokens relates to originality (relative to the training distribution), and **figure 8** also shows that 'common' images have lower reconstruction error at any number of learned (inverted) tokens than 'original' images.

**Weaknesses:**

**Originality is defined w.r.t. training data**. I think this particularly problematic. An artist can have a very unique/original style, but if their work is very well represented in the training data, then your method would state that the artist's work is not original (as a single token reconstruction would probably be strong). It seems like your method more so measures a T2I model's familiarity/ease of replication for a given concept/image, which directly relates to how frequently the concept is seen during training, but not with how unique the concept is.

**Unclear how this can be applied to the real problem of copyright protection**. Would you use your measure to quantify how original generated images are? All experiments seem to measure the originality of real query images. Much remains to be done to show this measure can be useful in practice (e.g. how would a lawyer incorporate your method to argue for or against a case of copyright infringement?). I think claiming this work is built on legal notions is an overstatement.

**Heavy reliance on uninterpretable techinques, like textual inversion and DreamSim**. How do we know that textual inversion methods are equally effective for different kinds of images, and that DreamSim is equally reliable in measuring image similarity for images from different domains? Also, these tools are uninterpretable, which I fear limits their utility for a non-technical audience, like judges, juries, and lawyers who would ultimately engage in discussions of copyright.

**Limited experiments**. Especially for the real world case -- having only 20 images per concept and using a 'legal expert' to determine if an image is common or not is rather unscientific.

See questions.

Nit picks / typos:
- L72 "Specifically, We"
- L84 "findings underscore demonstrate"
- L125 "synthesis" --> synthesize
- "recreate unseen elements" (L34) "reconstruct novel elements" (184) feel contradictory
- "unimaginative reconstruction" (L85) the word 'unimaginative' does not feel very scientific -- "replicating seen elements" feels more appropriate
- Many instances of '' as double quote, should be `` for open double quote.
- L282 "However, The"
- L479 \cite used instead of \citep
- L480 "Freenes" freeness?

**Questions:**

In Sec 2, is it fair to call the prompted elements "unseen" when you are using a pretrained text encoder? I appreciate the choice to use BERT instead of CLIP as the text encoder, but I think there is still 'concept leakage' by using BERT.

In Figure 8, why is the reconstruction loss not monotonically decreasing w.r.t. number of tokens?

I am not sure how the 'in distribution assessments' play in to your analysis? There are no results for these analyses presented in the main text. Is the point simply to say that your model has not memorized the training images? The inclusion of these discussions in the main text is a bit confusing.

Your experiments measure the 'originality' of real query images. Don't we ultimately care about how 'original' generated images (e.g. to determine if copyright has been infringed)? It is unclear how the measure should be applied and interpreted in practice.

How many images do you use to perform textual inversion for a single concept? In L212-237, it appears like multiple images are used

---

> ### Author Response · Authors · 2024-11-14
>
> We sincerely thank the reviewers for their valuable feedback and insightful comments on our submission.
>
>
> Addressing the weaknesses:
> (1)  "Originality is defined w.r.t. training data..." -  We agree that an image can be highly unique or original while still being well-represented in existing work. Providing a comprehensive view of originality is beyond the scope of any single study. Our approach, which examines whether an image is easily reconstructed from previous concepts and works, is just one facet of originality assessment. We do not intend for our method to be used as the sole arbiter; rather, it offers a quantitative assessment for a common question in such cases: how novel and distinct is the new creation relative to existing works. This approach aligns with the spirit of the law in assessing originality and distinctiveness.
>
>
> (2) "Unclear how this can be applied to the real problem of copyright protection..." - Our intended use case is when questions of originality for a particular image or concept arise and require quantitative evaluation. For a review of how such methods can be applied in practice, we refer readers to “Copyright Regenerated: Harnessing GenAI to Measure Originality and Copyright Scope,” published in the Harvard Journal of Law & Technology. However, the scope of this paper is focused on the technical aspects and challenges involved in such assessments.
>
>
> (3) "Heavy reliance on uninterpretable techniques, like textual inversion and DreamSim..." - We agree that if the intention is to apply the method within a specific context or for certain “kinds of images,” then experiments demonstrating the method’s utility would be appropriate. In this paper, we provided such experiments across five different concepts to illustrate how this can be done effectively.
>
>
> (4) "Limited experiments...." - We respectfully disagree. We include 20 images for 5 experiments and provide results based on the entire dataset of 100 images. Additionally, we present confidence intervals, and the results are statistically significant. Relying on human experts for annotation is also an established and accepted practice in such studies.
>
>
>
> Regarding the questions:
>
> "In Sec 2,..." - We don’t see how there can be a concept leakage given that the text encoder had no access to images.
>
>
> "I am not sure how the 'in distribution assessments' ..." - As mentioned in L277-278, The goal of assessment is to validate that the generations are not a result of the model overfiting to the single image. A more elaborate discussion is provided in the supplementary in the synthetic settings where we conduct an ablation study.
>
>
> "Your experiments measure the 'originality'... " - The use case we have in mind is querying the originality of real query images to determine whether they have certain protected rights rather than verifying the originality of generated images (though this could also be a potential use case, it is not the focus of this paper).
>
>
> "How many images do you use..." - We use only one.

---

### Official Review · Reviewer_5Xxq · 2024-11-06

**Soundness:** 2
**Presentation:** 3
**Contribution:** 2
**Rating:** 5
**Confidence:** 4

**Summary:**

In this paper, the authors propose a framework to measure originality in diffusion models. Authors first demonstrate the hypothesis that the generalization of a concept is related to the proportion of the concept’s presence in training time in a synthetic data setting. The generalization of concept is measured by how well a model can reconstruct an image via an inverted prompt. The authors then extend this study to real world SD 1.4 model on 100 images. The authors claim the length of textual inverted prompt needed to get good “dreamsim” score represents how original image is.

**Strengths:**

1. Understanding originality in T2I generated images is an important and timely topic
2. The synthetic data experiments are well motivated and very well-done
3. Unlike previous memorization studies which need access to training data, the metric proposed in this paper is test time and could be quite useful in practice

**Weaknesses:**

1. The authors skipped to cite a large number of relevant memorization and prompt inversion literature. To name a few [1,2,3,4,5,6].
2. A major concern I have is the authors' definition of originality. There is a lot of subjectivity associated with this and I believe the authors are not quite rigorous with their definition and their motivation on using dreamsim score to capture this. If the dreamsim score captures how similar two images are close to each other in both semantic and syntactic sense but not exact, I’d say they are original. Even in images shown in Fig 7, the “house” case, even with 5 tokens and seemingly low dreamsim score, I still feel reconstruction is still original.
3. The authors failed to follow up the synthetic data claims with real world data experiments. The SD experiment is not well designed, and many other factors come into play with SD 1.4 such as memorization induced by text encoder itself (as observed in [1]) that can impact the length of inverted prompt. Also see Q4 on what could be a reasonable real-world followup in my opinion.
4. The dataset used in SD 1.4 experiment is very small and the authors showed results only on the earliest SD models. The claims might not extend to recent models like SD 3 with alternative architectures, training losses and cleaned-up training datasets.
5. Some contradictions to proposed framework might exist such as the case of “Mothers influence on her young hippo” from [1], how many tokens will be needed to recover this I wonder.  Also would be nice to run your method on this specific image and discuss how the results align with or challenge your framework.
6. The writing can be improved.

-----
References:
[1] - Somepalli, Gowthami, et al. "Understanding and mitigating copying in diffusion models." Advances in Neural Information Processing Systems 36 (2023): 47783-47803.
[2] - Wen, Yuxin, et al. "Detecting, explaining, and mitigating memorization in diffusion models." The Twelfth International Conference on Learning Representations. 2024.
[3] - Gu, Xiangming, et al. "On memorization in diffusion models." arXiv preprint arXiv:2310.02664 (2023).
[4] - Somepalli, Gowthami, et al. "Diffusion art or digital forgery? investigating data replication in diffusion models. 2023 IEEE." CVF Conference on Computer Vision and Pattern Recognition (CVPR). 2022.
[5] - Wen, Yuxin, et al. "Hard prompts made easy: Gradient-based discrete optimization for prompt tuning and discovery." Advances in Neural Information Processing Systems 36 (2024).
[6] - Hao, Yaru, et al. "Optimizing prompts for text-to-image generation." Advances in Neural Information Processing Systems 36 (2024).

**Questions:**

1. L91 - The connections to Scheffer paper is very hand wavy, can you please explain this a bit better
2. The choice of dreamsim over other semantic feature extractors like CLIP is a bit unclear. Would be nice to show experimentally that this choice is better in your use case rather than what’s claimed in Dreamsim paper.
3. Why did you stop at 5 tokens? What happens if we make it 20 (like you showed in teaser)? Can we get perfect reconstructions with higher number of tokens or is it diminishing returns?
4. Would be nice to see a real world extension to your synthetic study. For example something like “a dog T2I model” from scratch (with BERT encoder) where you control for breeds, poses, and backgrounds and show your generalization hypothesis holds there too.
5. L311 - why did you choose BERT in your synthetic data? I went over to appendix L70, and the explanation is again very hand-wavy, can you expand on this?
6. Why is there an increase in avg dreamsim score at token_size=4 in Fig 8? Any intuitions there?
7. In conclusion, the authors claim “..offering insights that inform copyright analysis and legal protection..”.  Would be nice to see some qualitative/quant results support this. Maybe take some cases from previous memorization [1] or unlearning literature [2]?

----
References:
[1] - Somepalli, Gowthami, et al. "Understanding and mitigating copying in diffusion models." Advances in Neural Information Processing Systems 36 (2023): 47783-47803.
[2] - Gandikota, Rohit, et al. "Erasing concepts from diffusion models." Proceedings of the IEEE/CVF International Conference on Computer Vision. 2023.

---

> ### Author Response · Authors · 2024-11-14
>
> Thank you for your thoughtful and constructive feedback on our paper. We appreciate the time and effort you dedicated to reviewing our work, as well as your comments and suggestions.
>
> With regards to the weaknesses:
>
> (1) We thank the reviewer for pointing out these references, and we will address them in the final version of the paper. We will take the time to review these works more closely. However, note that the aim of our work is not to identify memorized examples, nor to perform prompt inversion. Instead, we identify commonalities and assess originality in a given work from the perspective of the generative model.
>
> (2) We agree that there is considerable subjectivity involved, and we do not attempt to provide a rigorous definition at any point in the paper. While we use the DreamSim score to assess reconstruction quality, it is not integral to our algorithm and can be substituted with any similarity measure acceptable to users. For evaluating results on our real-world data, however, we found DreamSim to be a particularly useful metric.
>
> (3) We are uncertain about what the reviewer means here and on what basis they assess the experiment as poorly designed. We emphasize again that our method is not intended to identify memorized examples. While there are indeed interesting directions for follow-up work and additional experiments, we respectfully ask that the reviewer evaluate the paper on its current merits rather than on potential future work.
>
> (4) We conducted experiments on SD 1.4 due to computational constraints. In our experiments, we demonstrated that using SD 1.4, we can effectively distinguish common images from original images in a meaningful way. While it may be interesting to investigate whether other trained models share this property (and we see no reason they wouldn’t), this does not diminish the validity of our results.
>
> (5) Again, we thank the reviewer for the reference; however, our goal is not to identify memorized data. We expect that memorized data would require only a few tokens to recover and would appear to the model as "common" rather than original, which does not contradict our results. Note, that in the supplementary, we specifically elaborate on this topic and show generic images that did not appear in the training data are still detected as common by the method.
>
>
> Regarding the questions:
>
> (2) Please see the discussion and ablation study in the supplementary (section 7)
>
> (3) Our stopping criteria is due to computational constraints, we do have (and intend to add) further results for up to 7 tokens.
>
> (5) In the synthetic experiments we wanted a model that has no training that involves images (such as clip), to make sure the visual understanding comes from the model and the model alone. We will deepen this discussion in the next version of the paper.
>
> (6) That is a good question, we don’t know the answer and we are also thinking about that. However, given our method there is no apriori argument for the curve to be monotone.
>
> (7) Thank you once again for the references. We will take the time to review them thoroughly and will include an appropriate discussion in the revised paper.

---

### Note · Authors · 2024-11-14

I have read and agree with the venue's withdrawal policy on behalf of myself and my co-authors.